# *In vivo* osseointegration evaluation of implants coated with nanostructured hydroxyapatite in low density bone

**Daniel Almeida**[1]*, **Suelen Cristina Sartoretto**[2], **Jose de Albuquerque Calasans-Maia**[3], **Bruna Ghiraldini**[4], **Fabio Jose Barbosa Bezerra**[4], **Jose Mauro Granjeiro**[5], **Mônica Diuana Calasans-Maia**[5]

1 Dentistry School, Universidade Federal Fluminense, Niteroi, RJ, Brazil, 2 Department of Oral Surgery, Dentistry School, Universidade Federal Fluminense, Niteroi, RJ, Brazil, 3 Orthodontics Department, Dentistry School, Universidade Federal Fluminense, Niterói, RJ, Brazil, 4 Dental Research Division, Dentistry School, Universidade Paulista, São Paulo, SP, Brazil, 5 Clinical Research Laboratory, Dentistry School, Universidade Federal Fluminense, Niterói, RJ, Brazil

* drdanieldealmeida@gmail.com

**Data Availability Statement:** All relevant data are within the paper and its Supporting Information files.

## Abstract

### Objective

This *in vivo* study, aimed to biomechanically, histomorphometrically and histologically evaluate an implant surface coated with nanostructured hydroxyapatite using the wet chemical process (biomimetic deposition of calcium phosphate coating) when compared to a dual acid-etching surface.

### Material and methods

Ten sheep (2–4 years old) received 20 implants, 10 with nanostructured hydroxyapatite coating (HAnano), and 10 with dual acid-etching surface (DAA). The surfaces were characterized with scanning electron microscopy and energy dispersive spectroscopy; insertion torque values and resonance frequency analysis were measured to evaluate the primary stability of the implants. Bone-implant contact (BIC) and bone area fraction occupancy (BAFo) were evaluated 14 and 28 days after implant installation.

### Results

The HAnano and DAA groups showed no significant difference in insertion torque and resonance frequency analysis. The BIC and BAFo values increased significantly (p<0.05) over the experimental periods in both groups. This event was also observed in BIC value of HAnano group. The HAnano surface showed superior results compared to DAA after 28 days (BAFo, p = 0.007; BIC, p = 0.01).

**Funding:** This study was financed by S.I.N. Implant System, Sao Paulo, Brazil, but the company had no influence in the design, execution, and analysis of the results. The authors report no other potential conflicts of interest for this work. - Initials of the authors who received each award Answer: M.D.C.M. - Grant numbers awarded to each author Answer: 01 (M.D.C.M) - The full name of each funder Answer: SIN Implant System. URL of each funder website Answer: https://www.sinimplantsystem.com.br/ Did the sponsors or funders play any role in the study design, data collection and analysis, decision to publish, or preparation of the manuscript? Answer: NO - The funders had no role in study design, data collection and analysis, decision to publish, or preparation of the manuscript.

## Conclusion

The results suggest that the HAnano surface favors bone formation when compared to the DAA surface after 28 days in low-density bone in sheep.

## Introduction

Commercial pure titanium (Ti) and its alloys are biocompatible materials that facilitate the osseointegration of dental implants, ensure tissue healing without foreign body reactions, and confer favorable responses from adjacent cell populations [1]. However, with the evolution of surface treatments, the machined or minimally roughened Ti surfaces of dental implants have been replaced to ensure greater predictability, increase the survival rates of dental implants, and accelerate the osseointegration process [2–5], particularly in low density bone such as the posterior maxilla [6].

The long-term success and stability of a dental implant are increased by the interaction of several factors, which can be classified as non-implant-related or implant-related. Non-implant-related factors include poor patient health, the surgical technique used, surgeon experience, and loading conditions [7]. Implant-related factors refer mainly to the macrogeometry of the implant and its topographical characteristics at the micro- or nano-scale obtained by surface treatment [8].

Dental implants should have a suitable combination of mechanical and biological properties, but it is challenging for a single material to possess all the desired properties. Surface modifications combine the beneficial properties of different materials to overcome this difficulty. Physical and chemical surface modification techniques can generally be divided into three groups: (1) adding materials with desirable functions to the surface (e.g., plasma-spray coating, physical vapor deposition, biomimetic deposition of calcium phosphate coatings, and surface immobilization of functional molecules); (2) converting the existing surface to more desirable compositions and topographies (e.g., ion implantation and electrochemical oxidations); (3) removing material from the existing surface to create specific topographies (e.g., grit blasting and acidic etchings). Changes in the implant surface that favor and anticipate the osseointegration period will be important for early implant loading and restoring patient masticatory function [9].

Hydroxyapatite (HA) is widely used as a biomaterial to fill bone defects and to coat the metal components of prostheses and dental implants. Synthetic HA is a well-known implant material with excellent biocompatibility characteristics, including non-toxicity, low biodegradability, bone affinity, and osteoconductivity due to its ability to strongly bond with natural bone tissue [10]. The rationale is that because of their chemical similarities, bone tissue may not recognize HA as foreign. Consequently, it may heal faster and integrate with HA-coated implants more firmly and coating metallic implants with HA appears to be a consistent choice.

Early HA coatings applied with the plasma spraying method were relatively thick and porous, and their uneven structure and low-bonding strength have been responsible for several clinical failures [11]. Many studies have explored approaches for overcoming the shortcomings of HA plasma spray coating.

The nanosized crystalline HA (HAnano) coating using the wet chemical process (biomimetic calcium phosphate deposition) is simple and cost-effective, with highly reproducible coating layer thickness and chemical composition. After applying the HAnano surface-coating liquid to the implant and spinning it, a short heat treatment is performed, which does not affect the substrate, and is significantly shorter compared to traditional coating procedures,

such as chemical vapor deposition (CVD) and thermal spray deposition (TSD). The result is a thin and homogeneous HAnano layer.

This study compared the primary stability and biological response of implants with biomimetic deposition of calcium phosphate coating (HAnano) and dual acid-etched (DAA) through resonance frequency and histomorphometric analyses of bone-implant contact (BIC) and bone area fraction occupancy (BAFo) in low-density bone in sheep.

## Material and methods

### Material

This study used 20 dental implants 3.5 mm in diameter and 10 mm in length with conical macrogeometry, double inverted support threads, compact in the cervical region and body, and cutting at the apex divided into two groups of 10: the HAnano (Epikut Plus; S.I.N. Implant System; Sao Paulo, Brazil) group comprised implants with a nanostructured crystalline HA coating, and the DAA (Epikut; S.I.N. Implant System, Sao Paulo, Brazil) group comprised of implants with surfaces treated with DAA.

### Surface characterization

**Scanning electron microscopy (SEM).**   High-resolution SEM images obtained with FEI-QUANTA 450 (Thermo Fisher Scientific; Waltham, MA, USA) were used to examine the implant's surface topography (one sample per group) at an accelerating voltage of 10 kV, focal width of 3.0, and magnifications of 3000× and 15,000×.

**X-ray photoelectron spectroscopy (XPS).**   An X-ray energy dispersive spectroscopy analysis was used to determine the chemical composition of each surface with an accelerating voltage of 20 kV and focal width of 40 using an EDAX detector equipped with a dual beam electron microscope (AMETEK Materials Analysis Division; Mahwah, NJ, USA) and the Genesis software program (EDAX LLC; Mahwah, NJ, USA).

### *In vivo* analysis

**Animals.**   This study adhered to the guidelines of the Animal Research: Reporting of *In Vivo* Experiments [12] and Planning Research and Experimental Procedures on Animals: Recommendations for Excellence [13]. Inclusion criteria would be healthy sheep, aged 2 to 4 years, weighing 30 to 45Kg. Any sheep showing disease or not meeting the age and weight criteria would be excluded from the study. We used ten adult Santa Ines sheep with a mean age of three years (range: 2–4 years) and a mean weight of 37.05 kg (31–42 kg). To minimize the effects of subjective bias when allocating animals to treatment, they were randomly allocated to two experimental periods of two and four weeks using the coin toss method.

All procedures were performed in accordance with the National Institutes of Health (NIH) guide for the care and use of laboratory animals [14] and Brazilian guidelines for the care and use of animals in teaching or scientific research activities (DBCA) of the National Council for the Control of Animal Experimentation [15]. A veterinarian with over 20 years of experience conducted the nutritional care, fasting, and pre- and post-operative care of animals. The study protocol was approved by The Ethics Committee on Animal Use of the Universidade Federal Fluminense (No. 9.531.061.119). All experiments were conducted between March and July 2020. No animals were euthanized at the end of the study in accordance with the guidelines of the Reduction, Refinement, and Replacement (3Rs) Program, whose goal is to reduce the number of animals used in experiments, minimize their pain and discomfort, and avoid

euthanasia at their end [16]. All animals were immunized against common sheep diseases and were monitored for good physical condition.

The animals were housed at the Universidade Federal Fluminense's farm in a semi-extensive fenced system with native forage and brachiaria grass (*Brachiaria humidicola* and *Brachiaria decumbens*). In the preoperative period, animals received feed composed of the pastures, and in the postoperative period, pastures with nutritional supplementation for sheep and mineral water *ad libitum*. To reduce the preoperative stress levels of animals, the veterinarian transferred the sheep from the farm to the research center two weeks before surgery to allow for proper acclimatization [17]. The animals fasted for eight hours before surgery.

**Sample size calculation.** We used the data of a previous study [18] for the sample size calculation with the online platform Sealed Envelope (https://www.sealedenvelope.com/power/continuous-superiority/), considering a superiority study between HAnano- and HA-coated implants. In that study, the primary endpoint of BIC after 28 days found 66.5% and 76.5% for the experimental and control groups, respectively. Considering a significant 15% BIC effect after 28 days, the calculated sample size was five animals per group at a 5% significance level and 80% power (1-beta).

**Surgical model.** We used the sheep iliac crest animal model, consistent with previous studies [4, 5, 19–21]. Two implants were installed in the cranial part of the right iliac crest of each animal (n = 5/group; **Fig 1**).

**Anesthesia and analgesia.** The animals were pre-medicated intravenously with 0.05 mg/kg of acepromazine (Acepran; Vetnil; Louveira, Sao Paulo, Brazil) and 0.2 mg/kg of Diazepam (Teuto; Anapolis, Goias, Brazil) and intramuscularly with 0.4 mg/kg of morphine (Dimorf; Cristalia; Itapira, Sao Paulo, Brazil). After 20 minutes, the animals were unresponsive to pain, and cannulation of the cephalic vein was initiated with the intravenous administration of 5 mL/kg/h of Ringer's solution with lactate (Baxter Hospitalar LTDA; Sao Paulo, Brazil). Anesthesia was induced intravenously with 4 mg/kg of propofol (Baxter Hospitalar LTDA) and maintained with 1% isoflurane (Cristalia) after orotracheal intubation. In addition, 4mg/kg of lidocaine (Xylestesin; Cristalia) and 0.1 mg/kg of morphine (Dimorf; Cristalia) was used as an epidural block. An experienced veterinary anesthesiologist administered the anesthesia and supervised all procedures during surgery.

**Surgical procedure.** The right side of the iliac crest was initially trichotomized with a razor blade to enable the application of a 0.5% chlorhexidine antiseptic solution. An approximately 5 cm incision was made in the iliac crest region of the animal with a n˚ 3 scalpel handle (Bard Parker; Aspen Surgical; Caledonia, MI, USA) and n˚ 15 blade (Solidor; Lamelid; Osasco, São Paulo, Brazil). After incision, skin, muscle, and periosteum were detached to expose the skeletal plane. The manufacturer recommended drilling sequence was used with low rotation (1200 rpm) and abundant irrigation using 0.9% sodium chloride solution (Darrow Laboratorios AS; Rio de Janeiro, Brazil) to avoid tissue necrosis by overheating.

Two implants were installed with the aid of a contra-angle coupled to a surgical electric micromotor (BLM 600 Plus; K Driller; Sao Paulo, Brazil). The implants were installed at 24 rpm in the first ewe after randomly selecting the position of each one using the sealed envelope method. In the next ewe, the implant position was rotated clockwise, and both implants were installed in different locations. Consequently, no implant was placed at the same site as another. The inter-implant distance was at least 5 mm, and all were positioned equicrestally. The insertion torque value (ITV) was documented for each implant according to the drilling unit. For ITV >50 N/cm, an analogic wrench (S.I.N. Implant System) was used for measurement.

A single operator performed the surgeries, and immediately after implant installation, resonance frequency analysis (RFA) was performed using two different tools: Osstell with

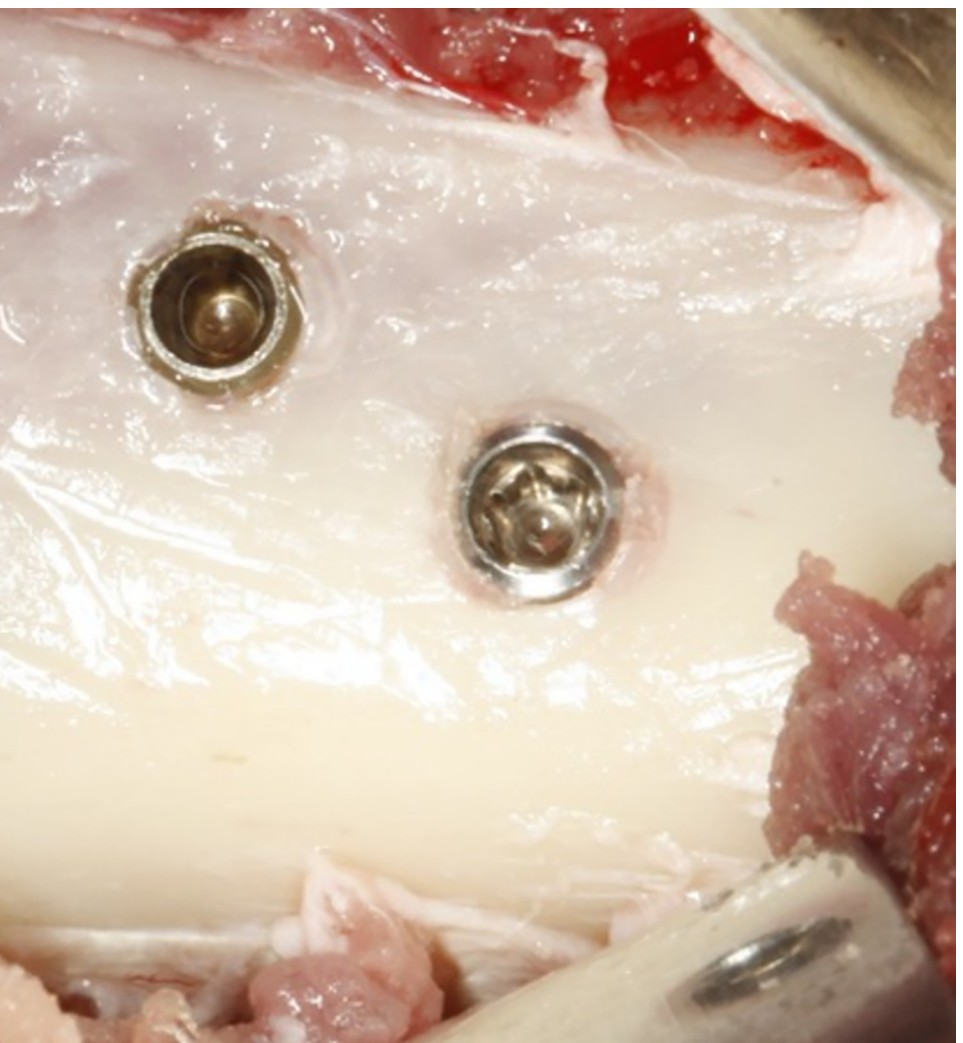

**Fig 1.** Surgical procedures to install the implants: (A) Installation of implants in the iliac crest while maintaining the safety distance between them (5 mm). (B) Removal the bone blocks with the implants after 2 and 4 weeks using a trephine drill with a 5mm internal diameter. (C) The iliac crest region after removal of the bone block with the implant. (D) Trephined bone block with the implant.

SmartPeg magnetic transducers (Integration Diagnostics; Savedalen, Sweden) and Penguin with MulTipeg (Penguin, Integration Diagnostics; Goteborg, Sweden). Finally, absorbable sutures with 4–0 Vicryl thread (Ethicon, Inc.; Somerville, NJ, USA) were used to close the repositioned periosteal flap and skin sutures with 5–0 nylon thread (Ethicon, Inc.). The operative wounds were left uncovered, and the surgery region received no external immobilization.

**Implant stability measurements.** Surgeons performed RFA immediately after implant installation with the Osstell implant stability quotient (ISQ) and Penguin RFA devices. The MulTipeg magnetic transducers were mounted on each implant and hand-tightened using the manufacturer's recommended metal key. The Penguin RFA probe was held 1 mm from the MulTipeg, and the ISQ was recorded on the digital instrument display for each implant. Three consecutive measurements were recorded in the lateral direction. In this study, the final ISQ value of each implant was the average of the three measured ISQ values. In addition, RFA measurements were collected with the Osstell ISQ device using the SmartPeg system and the plastic

key provided by the manufacturer. The average of the three ISQ values was taken as the final ISQ for each implant.

**Post-operative case.** All animals received antibiotic therapy by intramuscular injection of 0.1 ml/kg of oxytetracycline (Terramycin; Pfizer; New York, NY, USA) every 48 hours for three days. In addition, 4 mg/kg of Tramal (Pfizer) and 0.5 mg/kg of anti-inflammatory meloxicam (Meloxivet; Duprat; Rio de Janeiro, Brazil) were administered daily over five days. Oxytetracycline spray with hydrocortisone was used daily at the wound site (Terra-Cortril Spray; Zoetis; Sao Paulo, Brazil). Zinc oxide ointment with cresylic acid (Unguento Chemitec; Chemitec; Sao Paulo, Brazil) was applied along with silver spray (Aerocid Total; Agener Uniao; Aracoiaba da Serra, Brazil) to promote healing and deter insects.

**Histological processing.** After two and four weeks, the animals were reoperated for implant removal with a 5-mm trephine drill (S.I.N. Implant System). The surgical procedures were as described above, and all the sheep were subsequently returned to the farm, where they recovered completely. Samples containing bone and implant were fixed in 4% buffered formaldehyde solution for 48 hours and then dehydrated in increasing alcohol solutions of 60%, 70%, 90%, and 100%. Next, infiltration with light-curing resin (Technovit 7200; Kulzer & Co.; Wehrheim, Germany) was performed according to the manufacturer's instructions. Then, the samples were embedded in the same resin, cut in the apical-coronal plane using a macro-scale cutting and grinding technique (Exakt 310 CP series; Exakt Apparatebau; Norderstedt, Germany), and sanded and polished to a final thickness of 30 to 40 μm. The slides were stained with toluidine blue to identify the bone tissue and acid fuchsin for background contrast. Light microscopy at 10× and 20× magnification (Olympus BX43; Olympus Corporation; Tokyo, Japan) was used to analyze the sections, with images acquired using the cellSens software (Olympus Corporation) and polarization microscopy (Axioplan; Carl Zeiss AG; Oberkochen, Germany) allowing visualization of the general orientation of bone collagen fibers.

## Histomorphometric analysis

Photomicrographs at 10× magnification were captured in sequenced fields from each histologically processed slide (Fig 2) to scan and reconstruct the total area of the implant and adjacent bone (Fig 2A). After reconstruction of all images, the area of interest was determined and drawn vertically from the first implant thread to the beginning of the fourth thread (Fig 2B). This vertical delimitation was used to determine the BIC value, which was transformed into a percentage. The implant profile design was then duplicated and aligned at 270 μm in the horizontal plane, completing the total area of interest (Fig 2C). The BAFo score was manually determined with Image J software (NIH; Bethesda, MD, USA) and transformed into a percentage (Fig 2C). All histologic slides were coded according to the experimental groups and periods, and two experienced examiners blindly evaluate the slides.

## Statistical analysis

The insertion torque and ISQ values using Osstell and Penguin devices did not pass the Shapiro-Wilk normality test. Consequently, their log-transformed values in the DAA and HAnano groups were compared using a Student's t-test with a significance threshold of $p < 0.05$.

The histomorphometric BIC and BAFo values in the DAA and HAnano groups are reported as mean ± 95% confidence interval (CI) across five animals per group and experimental period. These values passed the Shapiro-Wilk normality test and were compared using a Student's t-test with a significance threshold of $p < 0.05$ to investigate between surfaces and experimental periods.

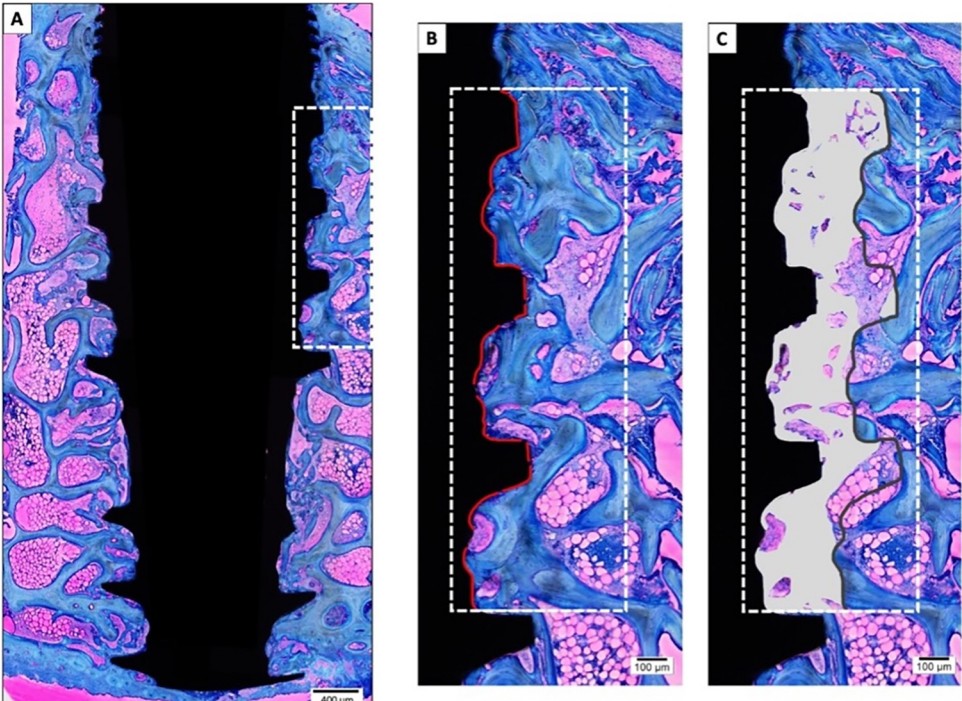

**Fig 2. Histomorphometry for BIC and BAFo analysis. (A)** Histological reconstruction of the implant and adjacent bone. **(B)** The line of interest for BIC evaluation is shown within the dotted box. In the long axis of the implant, the profile drawing was traced from the first implant thread to the beginning of the fourth thread. The red line denotes BIC. **(C)** Delineation of the BAFo area of interest. A line identical to the implant profile diagram was duplicated and aligned at 270 μm in the horizontal plane (total area). BAFo was manually determined for further analysis (total area/ BAFo; %). Staining with toluidine blue and acid fuchsin was used. Scale bars: 400 μm, A; 100 μm, B and C.

All statistical analyses were performed using the GraphPad Prism v.8.3 software (La Jolla, CA, USA).

## Results

### Surface characterization

**SEM.** SEM visualization of implant surfaces showing textured microstructures and topography (**Fig 3**). The surface textures observed at intermediate (3,000×) and high (15,000×) magnifications showed considerable similarity in surface morphology between the HAnano (**Fig 3A and 3B**) and DAA (**Fig 3C and 3D**) implant groups.

**XPS.** XPS showed the presence of calcium only in the HAnano group, while Ti, vanadium, and phosphorus peaks were observed in both groups (**S1 Fig**).

**ITV.** ITV did not differ significantly between the HAnano and DAA groups ($p > 0.05$), with both groups having an average ITV of 72–74 N/cm (**Fig 4**; **S1 Table**).

**RFA.** RFA was performed with the Osstell/SmartPeg and Penguin/MultiPeg devices and transducers. The log transformed ISQ values of the HAnano and DAA groups are shown in **Fig 5**. Penguin/MultiPeg showed significantly higher resonance frequencies than Osstell/ SmartPeg for all surfaces ($p < 0.05$). However, ISQ did not differ significantly between surfaces with the same device ($p > 0.05$; **S1 Table**).

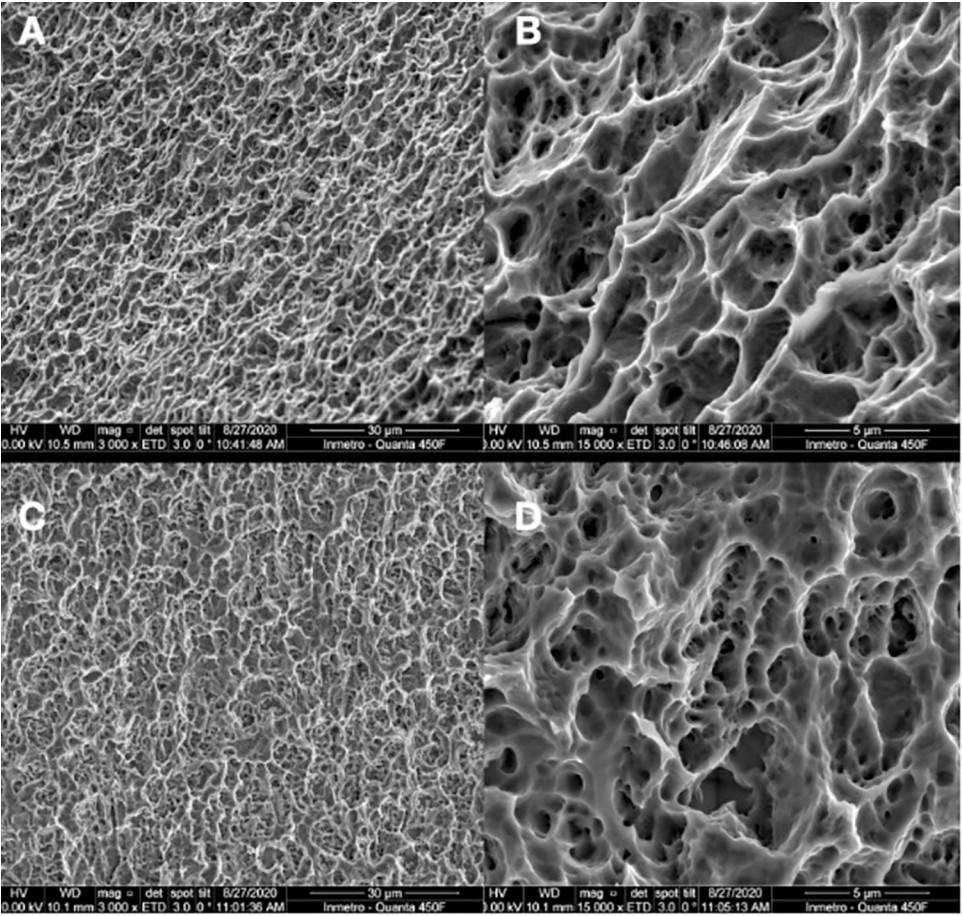

**Fig 3.** SEM photomicrographs of the surface of (**A+B**) HAnano and (**C+D**) DAA implants at 3000× (scale bar = 30 μm; A+C) and 15,000× (scale bar = 5 μm; B+D) magnification.

### *In vivo* study

The anesthesia, surgical intervention, and implant installation were uneventful. All animals recovered quickly and were cleared to walk after surgery. All animals gained weight after surgery except one that lost almost 10% of its initial weight 28 days after surgery. However, no changes were observed in histological findings of this animal, so the weight loss likely did not have biological effects. No cases of superficial or deep infection occurred. No implants showed clinical mobility, bone loss, or infection.

### Histological evaluation

A descriptive microscopic evaluation of non-decalcification enabled a qualitative assessment of the biological response to the tested surfaces. Two weeks post-surgery, both HAnano and DAA surfaces showed peri-implant bone healing between threads. The HAnano coating showed more compact trabecular bone between the implant threads (**Fig 6C and 6D**) than the DAA group (**Fig 6A and 6B**). Several BIC regions and scarce islands of bone debris in contact with trabecular bone were observed in both groups.

At 28 days post-implantation, DAA (**Fig 6E and 6F**) and HAnano (**Fig 6G and 6H**) surfaces show a time-dependent increase in newly formed bone volume adjacent to the implant surface

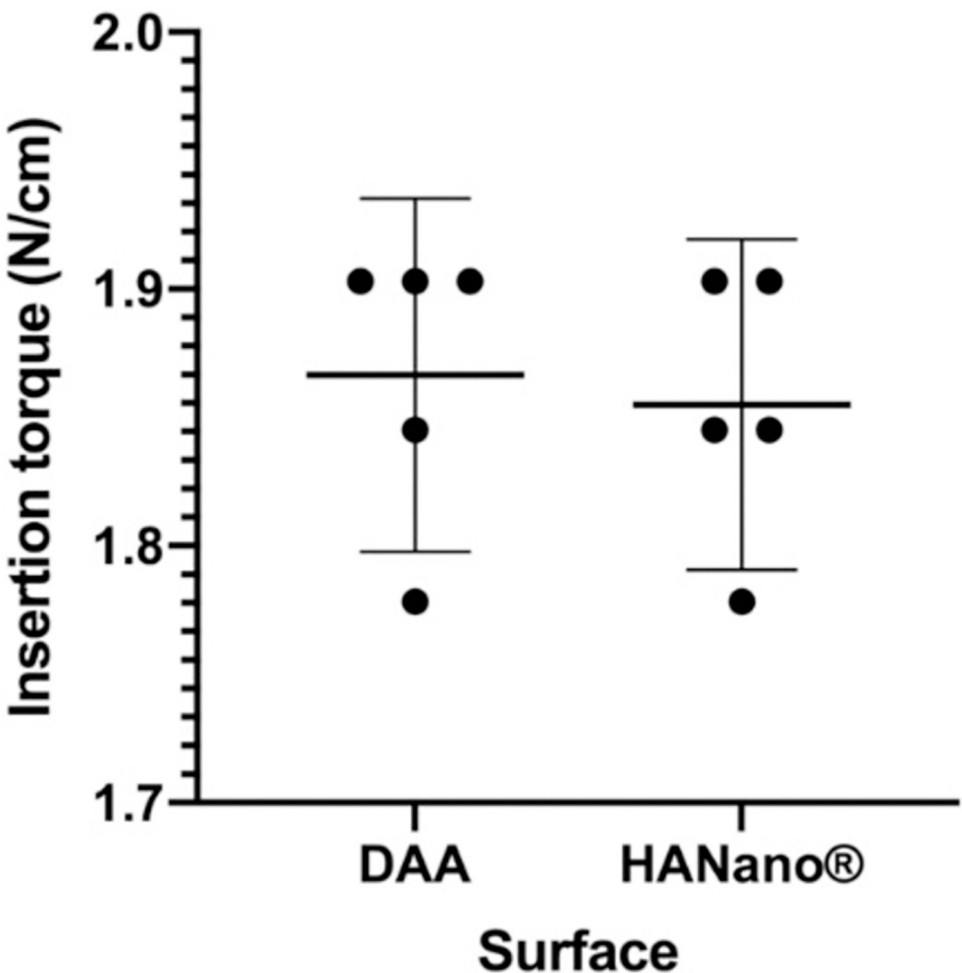

**Fig 4. ITVs (N/cm) of HAnano and DAA surfaces.** The graph shows the distribution of all point values as the mean and 95% CI across five samples.

compared to the previous time point. The HAnano group showed a greater increase in bone trabecular structure than the DAA group, with intimate contact between the newly formed bone and the implant surface. While both HAnano and DAA surfaces showed large BIC areas, the HAnano surfaces showed an intimate and nearly complete BIC.

**BIC.** BIC did not differ significantly between the DAA (56.76, 95% CI: 40.22–73.31) and HAnano (66.09, 95% CI: 49.80–82.37) groups 14 days post-implantation. However, BIC increased significantly in a time-dependent manner in the HAnano group ($p = 0.02$), surpassing 80% by 28 days (82.27, 95% CI: 78.08–86.47). Moreover, BIC was significantly higher in the HAnano group than in the DAA group (71.05, 95% CI: 62.26–79.85; $p = 0.01$) after 28 days (**Fig 7**; **S2 Table**).

**BAFo.** Similar to BIC, BAFo did not differ significantly between the DAA (40.04, 95% CI: 31.09–48.99) and HAnano (47.96; CI 41.29–54.64) groups 28 days post-implantation. However, BAFo increased significantly in both groups in a time-dependent manner compared to the previous period ($p = 0.007$). In addition, BAFo was significantly higher in the HAnano group (65.53, 95% CI: 57.80–73.27) compared to the DAA group (54.31, 95% CI: 50.18–58.45; $p = 0.007$), occupying >60% of the analyzed area (**Fig 8**; **S2 Table**).

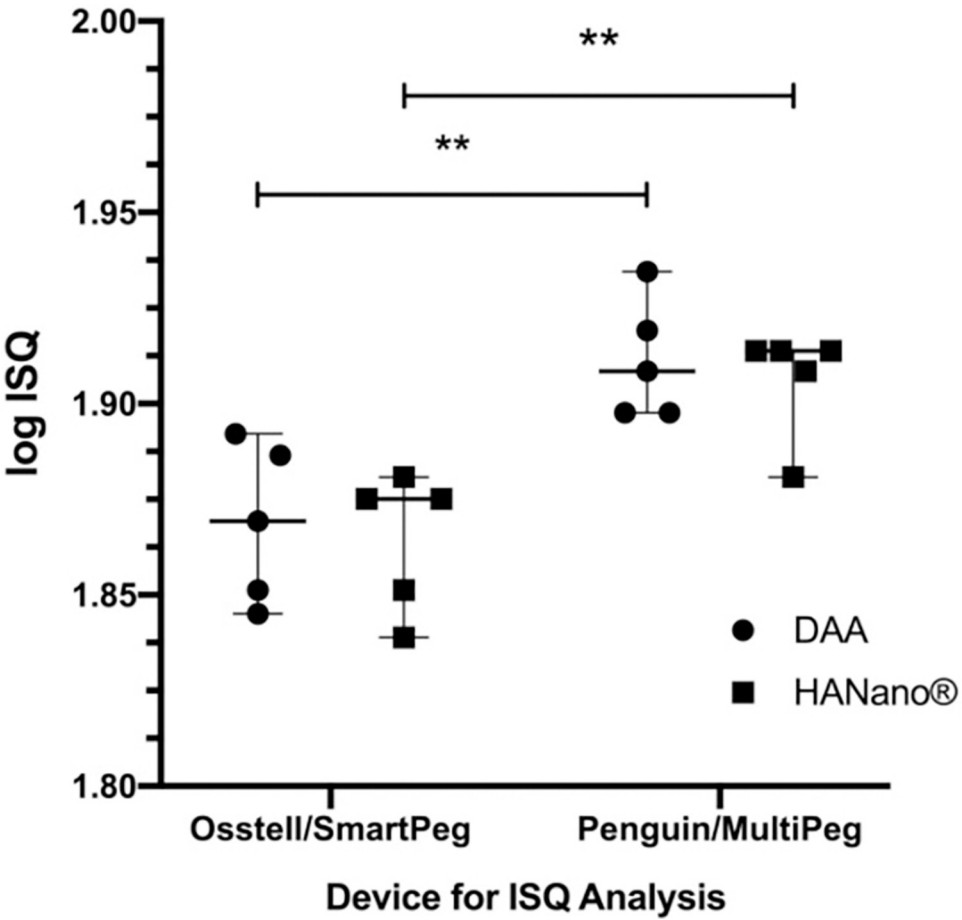

**Fig 5. ISQ values of HAnano and DAA surfaces.** The graph shows the distribution of ISQ values for the HAnano and DAA surfaces measured using two device/transducer pairs across five samples per group. Key: **, $0.002 \leq p \leq 0.003$.

## Discussion

This *in vivo* study on low-density bone used the sheep iliac crest model for implantation, which is suitable for biomedical research due to its similarities to humans in weight, joint structure, bone tissue, and bone regeneration [16, 22] and advantages over other experimental animal models in bone composition, metabolism, remodeling, and regeneration time [22]. Previous studies of dental implant designs and surfaces have used sheep [19–21, 23, 24], rabbit [7, 25, 26], and dog [27–29] models. The most commonly used sites in these animals are the mandible [27, 28] and tibia [29] in dogs, the tibia [4, 7, 25, 26] in rabbits, and the tibia [5, 23], iliac crest [19–21, 24], and mandible [30] in sheep. The iliac crest site offers advantages in bone characteristics, lower postoperative morbidity, and the number of implant tests that can be performed simultaneously. However, while the iliac crest has low bone density, its cortical is thicker than that of the posterior maxilla region, which also has low bone density. The animal model we selected was considered an excellent alternative because it allowed for more samples per animal and the use of real dental implants, not prototypes, in the analyses.

Previous studies using a sheep model sacrificed the animals at the end of their experiments [19–21, 24]. However, in this study, all animals were kept alive and healthy after its conclusion. In addition, in this study, all implants were installed at 1,200 rpm and under copious irrigation

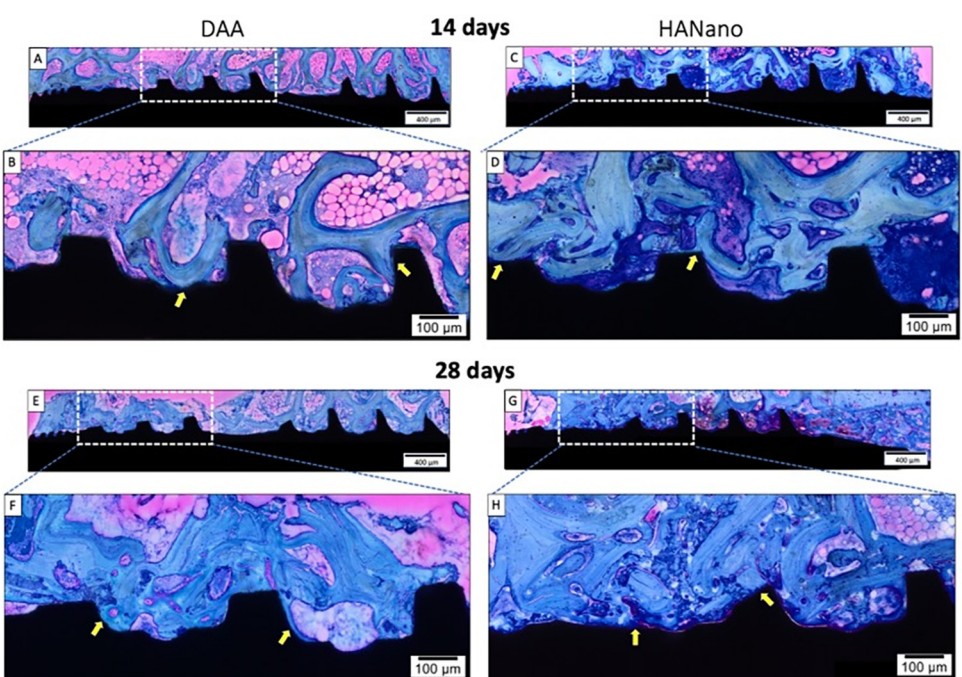

**Fig 6. Representative photomicrographs of the implants at 14- and 28-days post-implantation.** The dashed area is the square magnification. **(A+B)** DAA and **(C+D)** HAnano groups 14 days post-implantation. **(E+F)** DAA and **(G+H)** HAnano groups 28 days post-implantation. Staining used toluidine blue and acid fuchsin. Scale bar: 400 μm, A, C, E, and G; 100 μm, B, D, F, and H.

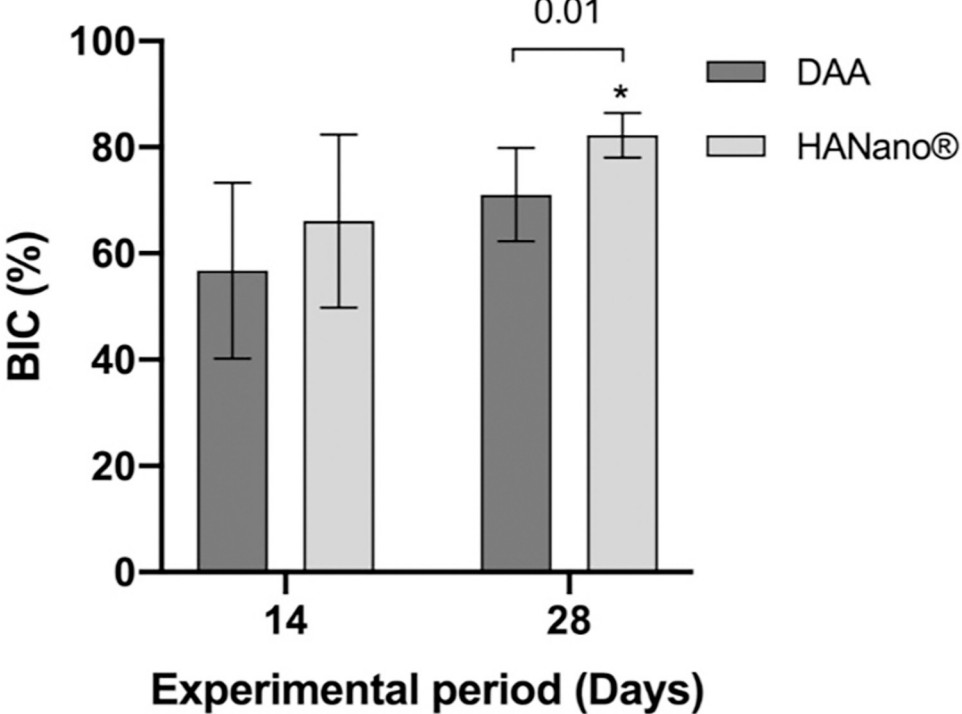

**Fig 7. Histomorphometric BIC (%) results as a function of the DAA and HAnano surfaces and 14- and 28-day experimental periods.** BIC values are presented as the mean with 95% CI across five samples. Key: *, $p = 0.02$.

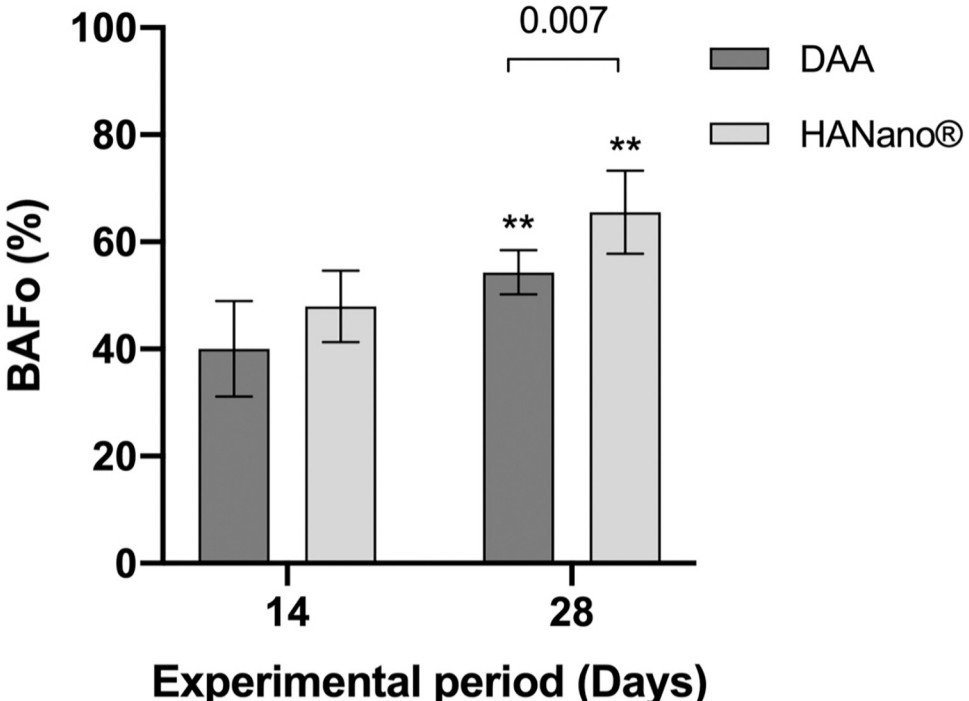

**Fig 8. Histomorphometric BAFo (%) results as a function of the DAA and HAnano surfaces and 14- and 28-day experimental periods.** BAFo values are presented as the mean and 95% CI across five samples. Key: **, 0.001≤p≤0.003.

with 0.9% saline solution, which is important for limiting bone heating. However, some previous studies did not report this in their methodologies [31, 32]. Moreover, it should be noted that, in this study, care was taken to rotate implant placement to avoid one group having more implants placed in a specific region of the iliac crest than another. A previous study with a similar experimental model evaluated BIC at the 14-, 28-, and 56-day periods [20], findings no significant differences between the 28- and 56-day periods. Therefore, this study only used the 14- and 28-day experimental periods.

The exploration of HA coating on implant surfaces was initially performed at a micrometric scale [33]. However, HA nanostructured surfaces were considered more promising for achieving rapid osseointegration in low-density bone, a challenge that persists in implant dentistry [34]. Therefore, using HA coatings on metallic implants should improve the speed of patient rehabilitation by decreasing the time from implant insertion to installation of the final prosthesis, particularly in regions of lower bone density [35]. The rough implant surfaces achieved by HA coatings offer osteoconductive benefits due to increased cell adhesion [36].

Surface interaction and changes at the implant-cell interface result in diverse cellular responses. Understanding the intracellular signaling involved in the adhesion, proliferation, and differentiation mechanisms of osteoblasts on dental implants is critical for successful osseointegration. The first steps in this implant response involve molecular adsorption and establishing an organic coating responsible for guiding the functions of surrounding cells, leading to the activation of specific genes [37]. A previous *in vitro* study showed that the HAnano blasted Ti surface improved wettability and made the implant surface super-hydrophilic [38]. This increased wettability facilitates the attachment process of bone cells to the implant surface, favoring osseointegration [39]. In addition, the HAnano surface has been

previously found to promote cell proliferation, viability, spreading, and secretion of collagen type I and osteopontin, favoring early osseointegration events [40].

In this study, we compared the biological performance of a new nanostructured HA coating, Epikut Plus, with a DAA surface, Epikut. Both study groups had implants with the same macrogeometry, varying only in the surface coating. Notably, this macrogeometry was developed especially for application in areas of low bone density presenting compact spirals. The biomechanical evaluations generally did not suggest any significant difference in osseointegration success between the two surfaces. However, histological evaluations showed a significant difference favoring the HAnano surface.

ITV is a direct measure of bone shear strength during implant installation surgery influenced by surgical bed preparation, implant design, and bone quality [41]. In this study, we confirmed the absence of nanostructured implant surface effects on insertion torque, consistent with a previous study [42]. All tested implants showed relatively high ITV of 60–80 N/cm, likely due to the strength of the thicker cortical bone of the iliac crest region and the implant's macrogeometry with compacting reverse-supported threads that favor insertion torque even in low-density bone regions. Another sheep-based study observed low ITV in all groups (<21 N/cm), but the implants used had a macrogeometry that promoted little bone compaction, which may have contributed to the low ITVs recorded [22]. Two systematic reviews found no correlation between insertion torque and osseointegration failure or marginal bone loss [43, 44]. In addition, another study in rabbit tibiae observed greater peri-implant bone formation in implants with higher (>50 N/cm) compared to lower (<10 N/cm) torque [45]. In this study, the implants used in the HAnano group, Epikut Plus, have a macrogeometry that allows for high insertion torque with immediate loading and nanostructured HA surface to support osseointegration. A synergistic effect of this combination has been previously reported [46].

Primary stability is defined as the absence of movement after intraosseous implant insertion. RFA is one of the most used methods to evaluate the primary stability of implants quantitatively. This analysis provides information about the stiffness of the bone-implant junction, with the results recorded as the ISQ. The findings of this study are consistent with previous randomized clinical trials that found acceptable primary stability with average ISQ values in humans [47]. ISQ values are directly affected by surgical instrumentation [48], bone density, implant size, and the macrogeometry of the implant body [49]. Since this study used the same implant model, implant size, and recipient area in both experimental groups, they had similar ISQ levels. A clinical study using tapered thread-compacted implants in the posterior maxilla reported a mean ISQ value of 53.66±12.04 [50], very close to our results.

This study evaluated the resonance frequencies immediately after implant installation using two different devices, Osstell and Penguin, and two magnetic transducers, SmartPeg and MulTipeg, respectively. Statistical differences were observed between them, with the combination of Penguin and MulTipeg providing the highest values. These results differ from other study [51], who evaluated primary stability using the same devices and transducers but found slightly superior results in their Osstell group. However, unlike this study, they used an *ex vivo* model.

In this study, resonance frequency was evaluated only on implant installation. We did not evaluate it in the later experimental periods to limit any damage caused by the insertion torque of the transducers in the implant and to avoid prejudicing the histological and histomorphometric evaluations of the bone-implant interface. A recent systematic review [52] found no correlation between insertion torque and primary stability, suggesting that a high insertion torque does not necessarily correspond to high ISQ. In this study, we recorded high levels of insertion torque and primary stability.

The BIC measures the direct relationship between bone tissue and implant. Our BIC results showed a steady increase in the trabecular bone adjacent to the implant surface from two to

four weeks, with a significant difference found between the HAnano groups at 14 and 28 days. Furthermore, the HAnano/28-day group showed a significant difference compared to the DAA/28-day group. These favorable results for the HAnano group are likely related to its nanostructured HA surface. Notably, several previous studies using nanostructured HA in animals support our BIC results [46, 53, 54].

BAFo increased from two to four weeks, with a significant difference between the HAnano and DAA groups. While a previous study showed comparable results between HAnano, SLActive (BLX; Straumann; Basel, Switzerland), and TiUnite (NobelActive; Nobel Biocare; Göteborg, Sweden) surfaces [20], the HAnano surface has not previously been compared with the DAA surface. HAnano implants are treated with DAA prior to receiving the nanostructured HA treatment. In this study, we observed the histomorphometric benefits of this additional treatment.

Our results are consistent with previous studies that evaluated HA-coated implants in rabbit tibiae after two weeks [55], in dog alveoli after four weeks [56], and in goat iliac crests after four weeks [57]. In these studies, the biomimetic HA coating shortened the healing period of implants by increasing bone-implant interaction. Our BIC and BAFo values in the HAnano group were similar to other study [57], who obtained values of 57.5±8.5% for BIC and 43.6 ±9% for BAFo in a study of goat iliac crests after four weeks. Other study [56] reported similar results to this study, with BAFo (44.94±17.69%) and BIC (77.28±11.22% after four weeks in dogs, compared to our values of 65.53±6.22% and 82.27±3.38, respectively.

## Conclusions

The HAnano surface favored osseointegration in the early stages of low-density bone repair compared to the DAA surface in the sheep model.

## Supporting information

**S1 Fig. Energy dispersive spectroscopy revealed the presence of calcium only in the HAnano® group (A).** Peaks of titanium, vanadium, and phosphorus were observed in both groups.
(TIF)

**S1 Table. Insertion torque and resonance frequency values of the DAA and HAnano® groups according to device.**
(DOCX)

**S2 Table. BAFo and BIC values of DAA and HAnano® groups according to experimental time.**
(DOCX)

## Acknowledgments

The authors wish to thank INMETRO, and the staff of the SEM LABORATORY for their invaluable contributions to this work. Also, authors gratefully thanks to CNPq (400030/2018-7) and INCT-regenera (465656/2014-5, http://www.inctregenera.org.br/)—supported by CNPq and Faperj) and to Faperj project number E-26/10.000981/2019-Network Nano/Saude.

## Author Contributions

**Conceptualization:** Mônica Diuana Calasans-Maia.

**Data curation:** Fabio Jose Barbosa Bezerra.

**Formal analysis:** Suelen Cristina Sartoretto.

**Investigation:** Jose de Albuquerque Calasans-Maia.

**Methodology:** Suelen Cristina Sartoretto.

**Resources:** Bruna Ghiraldini.

**Software:** Suelen Cristina Sartoretto.

**Supervision:** Jose Mauro Granjeiro.

**Validation:** Suelen Cristina Sartoretto.

**Visualization:** Jose Mauro Granjeiro.

**Writing – original draft:** Daniel Almeida.

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
