## [Decision Letter · Decision Letter 0]

24 Oct 2022

PONE-D-22-24153IN VIVO OSSEOINTEGRATION EVALUATION OF IMPLANTS COATED WITH NANOSTRUCTURED HYDROXYAPATITE IN LOW DENSITY BONEPLOS ONE

Dear Dr. Daniel Almeida,

Thank you for submitting your manuscript to PLOS ONE. After careful consideration, we feel that it has merit but does not fully meet PLOS ONE’s publication criteria as it currently stands. Therefore, we invite you to submit a revised version of the manuscript that addresses the points raised during the review process.

We look forward to receiving your revised manuscript.

Kind regards,

Wenguo Cui, Ph.D

Academic Editor

PLOS ONE

Journal Requirements:

2. Thank you for submitting the above manuscript to PLOS ONE. During our internal evaluation of the manuscript, we found significant text overlap between your submission and the following previously published works, some of which you are an author.

- https://www.dovepress.com/the-influence-of-nanostructured-hydroxyapatite-surface-in-the-early-st-peer-reviewed-fulltext-article-IJN

Please revise the manuscript to rephrase the duplicated text, cite your sources, and provide details as to how the current manuscript advances on previous work. Please note that further consideration is dependent on the submission of a manuscript that addresses these concerns about the overlap in text with published work.

Reviewers' comments:

Reviewer's Responses to Questions

**Comments to the Author**

1. Is the manuscript technically sound, and do the data support the conclusions?

Reviewer #1: Yes

Reviewer #2: Partly

2. Has the statistical analysis been performed appropriately and rigorously? 

Reviewer #1: Yes

Reviewer #2: Yes

3. Have the authors made all data underlying the findings in their manuscript fully available?

Reviewer #1: No

Reviewer #2: Yes

4. Is the manuscript presented in an intelligible fashion and written in standard English?

Reviewer #1: Yes

Reviewer #2: Yes

5. Review Comments to the Author

Reviewer #1: Thank you for the opportunity to review this manuscript. The authors conducted in vivo osseointegration evaluation of implants coated with HAnano /DAA in low-density bone in sheep, and concluded that HAnano surface is superior to DAA surface. There are a few issue that need to be addressed:

1) The reason for employing DAA as a control (gold standard?) should be provided in Introduction section.

2) Biocompatibility and safety evaluation should be carried out.

3) The preparation of the experimental surfaces should be provided in MATERIAL AND METHODS section.

4) The surface roughness is also important for in vivo osseointegration and should be determined quantitatively (e.g., AFM).

5) Although the authors have provided literature to support their standpoint, 4 weeks is not enough for completely osseointegration.

6) Figure 1: the figure caption does not match the figure.

7) Supplementary Figure 1 is missing and should be provided.

Reviewer #2: This manuscript, submitted by Daniel Costa Ferreira de Almeida, demonstrates that the HAnano surface favors bone formation compared to the DAA surface after 28 days in low-density bone in sheep. However, the data in the manuscript does not seem to be sufficient to support the authors' conclusion. Also, there seems to be a lack of validation for some key processes. The authors should address the following concerns.

1. In the part of the introduction, the authors summarized the causes of dental implant failure in detail and induced the methods of biomimetic deposition of calcium phosphate coating. However, the author does not explain why chose DAA as a compared group.

2. The manuscript does not state what material the authors constructed the nano-hydroxyapatite coating on. Is it titanium?

3. The author should explain in detail how to calculate the sample size.

4. The author mentioned that "Similar to BIC, BAFo did not differ significantly between the DAA (40.04, 95% CI: 31.09–48.99) and HAnano (47.96; CI 41.29 – 54.64) groups 28 days post-implantation." This part of the description does not match the data in Figure 8.

5. The note's contents in figure 1 are not reflected in the figure. Figure 1 does not look complete.

6. What is the difference between Osstell/SmartPeg and Penguin/MultiPeg？Please mark the transducer placement position in the picture and explain its working principle.

7. The authors mentioned that HA coating on the Ti surface increased its hydrophilicity and described that this mechanism might be involved in the study, but there was no hydrophilicity verification.

8. As an in vivo study, it lacks biosafety verification.

9. Lack of osteogenic validation of new bone tissue. Immunohistochemistry of osteogenic-related proteins is necessary.

10. No implant diagrams or model diagrams are available in the manuscript and should be supplemented.

11. The authors only recorded the insertion torque value during surgical implantation and did not perform mechanical tests on the models at 2 and 4 weeks later, such as push-out test and extract torque value.

6. PLOS authors have the option to publish the peer review history of their article (what does this mean?). If published, this will include your full peer review and any attached files.

Reviewer #1: No

Reviewer #2: No

---

## [Author Response · Author response to Decision Letter 0]

23 Jan 2023

To: Academic Editor of PLOS ONE Wenguo Cui

Manuscript: PONE-D-22-24153

Title: IN VIVO OSSEOINTEGRATION EVALUATION OF IMPLANTS COATED WITH NANOSTRUCTURED HYDROXYAPATITE IN LOW DENSITY BONE

Thank you for the attention you have given to our manuscript originally entitled “In vivo osseointegration evaluation of implants coated with nanostructured hydroxyapatite in low density bone”. Indeed, we appreciated the comments and criticism of the reviewer and the opportunity you have given us to submit a new revision of our reworked manuscript. 

The authors would like to acknowledge the effective and unbiased review of the manuscript, believing that introduction of the suggested alterations produced a manuscript with better editorial and scientific quality. 

Please find below the point-by-point the answers to reviewers’ comments:

Reviewer #1: 

Thank you for the opportunity to review this manuscript. The authors conducted in vivo osseointegration evaluation of implants coated with HAnano /DAA in low-density bone in sheep and concluded that HAnano surface is superior to DAA surface. There are a few issues that need to be addressed:

1) The reason for employing DAA as a control (gold standard?) should be provided in Introduction section.

Answer: The aim of this study was to evaluate the biological response to implants coated with a nanostructured hydroxyapatite surface, so we chose to use as control group an identical implant to the experimental one, but without the coating. The implant with the DAA surface is the same as the nanoHA implant without the coating. This information was included in the material and methods section.

2) Biocompatibility and safety evaluation should be carried out.

Answer: A previous in vitro study using the same surfaces tested in this in vivo study showed that the nanoHA surface promoted increased cell proliferation and viability when compared to the control group (DAA). In addition, increased cell spreading as well as type I collagen and osteopontin secretion were observed, favoring the early events of osseointegration (Martinez et al, 2017). Another previous in vitro study (Bezerra et al., 2017) showed that nanoHA surface promotes crucial intracellular signaling network responsible for cell adapting on the Ti-surface. This information about previous studies was included in the Introduction.

3) The preparation of the experimental surfaces should be provided in MATERIAL AND METHODS section.

Answer: The machined implant was produced from commercially pure titanium (Grade 4) cylindrical bars. During the machining process, the implants were inspected for their critical dimensional characteristics, shape/position, surface finish, and mechanical requirements. After that, they received automated pre-washing by Centrifugal Disc units, and a hygiene process carried out inside controlled rooms (Clean Room) in high-performance automated cleaning systems (Ultrasonic Cleaning Systems). Then, the DAA surface was obtained from the machined implant surface that received nitric acid baths, followed by sulfuric acid in a micro corrosion process. The nanosized crystalline HA (HAnano) coated the DAA surface using the wet chemical process (biomimetic calcium phosphate deposition). All those information was published previously (DOI: 10.1002/jbm.a.37052) and were included in the Material and Methods section.

4) The surface roughness is also important for in vivo osseointegration and should be determined quantitatively (e.g., AFM).

Answer: The qualitative and quantitative surface topography from the two groups was published previously (doi:10.3390/ma12050840). This information was included in the text.

5) Although the authors have provided literature to support their standpoint, 4 weeks is not enough for completely osseointegration.

Answer: The aim of this study was to investigate the early stages of osseointegration, so, therefore we used 4 and 8 weeks for a comparative evaluation.

6) Figure 1: the figure caption does not match the figure.

Answer: The authors have added a new Figure 1 and rewritten the caption of the new Figure 1.

7) Supplementary Figure 1 is missing and should be provided.

Answer: The supplementary Figure 1 was provided.

Reviewer #2: This manuscript, submitted by Daniel Costa Ferreira de Almeida, demonstrates that the HAnano surface favors bone formation compared to the DAA surface after 28 days in low-density bone in sheep. However, the data in the manuscript does not seem to be sufficient to support the authors' conclusion. Also, there seems to be a lack of validation for some key processes. The authors should address the following concerns.

1. In the part of the introduction, the authors summarized the causes of dental implant failure in detail and induced the methods of biomimetic deposition of calcium phosphate coating. However, the author does not explain why chose DAA as a compared group.

Answer: The aim of this study was to evaluate the biological response to implants coated with a nanostructured hydroxyapatite surface, so we chose to use as control group an identical implant to the experimental one, but without the coating. The implant with the DAA surface is the same as the nanoHA implant without the coating. This information was included in the material and methods section.

2. The manuscript does not state what material the authors constructed the nano-hydroxyapatite coating on. Is it titanium?

Answer: Yes, the study used machined implant produced from commercially pure titanium (Grade 4) cylindrical bars. The DAA surface was obtained from the machined implant surface that received nitric acid baths, followed by sulfuric acid in a micro corrosion process. The nanosized crystalline HA (HAnano) coated the DAA surface using the wet chemical process (biomimetic calcium phosphate deposition). All these information were included in the Material and Methods section.

3. The author should explain in detail how to calculate the sample size.

Answer: The sample size calculation was rewritten according to the reviewer's suggestion.

Revised text: 

“The sample size calculation was performed using the web site (https://www.sealedenvelope.com/power/continuous-superiority/) and based on a previous study (Sartoretto et al. 2020) which used the same experimental animal model and evaluated BIC and BAFo as primary endpoints. A significance level (alpha) of 0.05 and a power (1-beta) of 0.9 were applied. As a parameter for the calculation, the mean of BAFo 28 days post-surgery was used (65.53% � 6.22). Considering an SD of 6.22 and 20% of the superiority of the experimental group as clinical relevance, the mean outcome in the control group was 52,42%, and the sample size was five animals per group.”

4. The author mentioned that "Similar to BIC, BAFo did not differ significantly between the DAA (40.04, 95% CI: 31.09–48.99) and HAnano (47.96; CI 41.29 – 54.64) groups 28 days post-implantation." This part of the description does not match the data in Figure 8.

Answer: We appreciate the reviewer's observation. There was a typo. The sentence was corrected as follows: 

Revised text: 

“Similar to BIC, BAFo did not differ significantly between the DAA (40.04, 95% CI: 31.09–48.99) and HAnano (47.96; CI 41.29 – 54.64) groups 14 days post-implantation. However, BAFo increased significantly in both groups in a time-dependent manner compared to the previous period (p=0.007). In addition, BAFo was significantly higher in the HAnano group (65.53, 95% CI: 57.80–73.27) compared to the DAA group (54.31, 95% CI: 50.18–58.45; p=0.007), occupying >60% of the analyzed area (Figure 8; Supplementary Table 2)”

5. The note's contents in figure 1 are not reflected in the figure. Figure 1 does not look complete.

Answer: A new Figure 1 and caption were provided.

6. What is the difference between Osstell/SmartPeg and Penguin/MultiPeg？Please mark the transducer placement position in the picture and explain its working principle.

Answer: The Osstell/SmartPeg and Penguin/MultiPeg are commercial devices that evaluate the Resonance Frequency Analysis (RFA) which is one of the most used methods to quantitatively evaluate the primary stability of implants. This analysis provides information regarding the stiffness of the bone–implant union and the results are recorded as the implant stability quotient. 

The two commercially available devices were used in this study to (1) compare the ISQ results of the two devices and (2) provide robustness to the results of each surface tested.

The device pointer was positioned laterally and tangent to the correspondent transducer. In a triangle format, three measurements were taken with each device/transducer. This information was included in the material and methods section as follows: 

Revised text: 

“Surgeons performed RFA immediately after implant installation with the Osstell (Integration Diagnostics, Savedalen, Sweden) implant stability quotient (ISQ) and Penguin (Penguin Integration Diagnostics, Göteborg, Sweden) commercially available RFA devices. The MulTipeg magnetic transducer was mounted on each implant tested and hand-tightened using the manufacturer’s recommended metal key. The Penguin RFA probe was positioned 1 mm laterally and tangent to the correspondent transducer. In a triangle format, three measurements were recorded and the ISQ was recorded on the digital instrument display for each implant.

In addition, RFA measurements were collected with the Osstell ISQ device using the SmartPeg system and the plastic key provided by the manufacturer with the same protocol cited above. For Penguin and Ostell, the average of the three ISQ values was taken as the final ISQ for each implant.”

7. The authors mentioned that HA coating on the Ti surface increased its hydrophilicity and described that this mechanism might be involved in the study, but there was no hydrophilicity verification. 

Answer: The authors have mentioned in the Discussion section two previous studies that evaluated the same surface that we used in this study (DAA and HAnano): “A previous in vitro study showed that the HAnano blasted Ti surface improved wettability and made the implant surface super-hydrophilic (<4o; Bezerra et al., 2017). This increased wettability facilitates the attachment process of bone cells to the implant surface, favoring osseointegration (Nishimura et al., 2018).” 

In this study, wettability analysis was not performed because this evaluation has been previously published in Barbosa et al., 2017 (DOI 10.1007/s40430-016-0700-x).

8. As an in vivo study, it lacks biosafety verification.

Answer: All animals were subjected to clinical and ultrasonic evaluations and were free from reproductive or clinical disorders. The animals were managed in an intensive system, fed with chopped Napier grass (Pennisetum purpureum cv. Cameron) and 300 g/per animal/daily of concentrate (16% of crude protein), and free access to water and mineral salt (Ovinof-os, Tortuga, Sao Paulo, Brazil). This information was included in the Material and Methods section.

9. Lack of osteogenic validation of new bone tissue. Immunohistochemistry of osteogenic-related proteins is necessary.

Answer: In this study all the bone blocks containing the implants were embedded in Technovit resin. After sections (30-40 µm) we tried to perform immunohistochemical analysis, but the results were not as good as we usually get with paraffin-embedded samples with thinner sections (4-5 µm), so these results were not included in this study. 

10. No implant diagrams or model diagrams are available in the manuscript and should be supplemented.

Answer: A model diagram of the implants was included as “Supplementary Figure 2”.

11. The authors only recorded the insertion torque value during surgical implantation and did not perform mechanical tests on the models at 2 and 4 weeks later, such as push-out test and extract torque value.

Answer: As our main objective was to histologically and histomorphometrically evaluate BIC and BAFo, the mechanical tests at 2 and 4 weeks after surgery were not performed in order not to cause any interference at the implant-bone interface that could impair the histological analysis.

---

## [Decision Letter · Decision Letter 1]

8 Feb 2023

IN VIVO OSSEOINTEGRATION EVALUATION OF IMPLANTS COATED WITH NANOSTRUCTURED HYDROXYAPATITE IN LOW DENSITY BONE

PONE-D-22-24153R1

Dear Dr. Almeida,

We’re pleased to inform you that your manuscript has been judged scientifically suitable for publication and will be formally accepted for publication once it meets all outstanding technical requirements.

Kind regards,

Wenguo Cui, Ph.D

Academic Editor

PLOS ONE

Additional Editor Comments (optional):

Reviewers' comments:

Reviewer's Responses to Questions

**Comments to the Author**

1. If the authors have adequately addressed your comments raised in a previous round of review and you feel that this manuscript is now acceptable for publication, you may indicate that here to bypass the “Comments to the Author” section, enter your conflict of interest statement in the “Confidential to Editor” section, and submit your "Accept" recommendation.

Reviewer #1: All comments have been addressed

Reviewer #2: All comments have been addressed

2. Is the manuscript technically sound, and do the data support the conclusions?

Reviewer #1: Yes

Reviewer #2: Partly

3. Has the statistical analysis been performed appropriately and rigorously? 

Reviewer #1: Yes

Reviewer #2: Yes

4. Have the authors made all data underlying the findings in their manuscript fully available?

Reviewer #1: Yes

Reviewer #2: No

5. Is the manuscript presented in an intelligible fashion and written in standard English?

Reviewer #1: Yes

Reviewer #2: Yes

6. Review Comments to the Author

Reviewer #1: In this revised manuscript, the authors have carefully answered all the questions point-to-point from the reviewers, and I am pleased to recommend to accept this work as it is.

Reviewer #2: The author responded to all the reviewer's concerns. But where is Supplementary Figure 2？ Did the author miss it?

7. PLOS authors have the option to publish the peer review history of their article (what does this mean?). If published, this will include your full peer review and any attached files.

Reviewer #1: No

Reviewer #2: No

---

## [Editor Report · Acceptance letter]

13 Feb 2023

PONE-D-22-24153R1 

*IN VIVO* OSSEOINTEGRATION EVALUATION OF IMPLANTS COATED WITH NANOSTRUCTURED HYDROXYAPATITE IN LOW DENSITY BONE 

Dear Dr. Almeida:

I'm pleased to inform you that your manuscript has been deemed suitable for publication in PLOS ONE. Congratulations! Your manuscript is now with our production department. 

Kind regards, 

on behalf of

Professor Wenguo Cui 

Academic Editor

PLOS ONE